# Assessing the Technical Efficiency of Timber Production during the Transition from a Production-Oriented Management Model to a Multifunctional One: A Case from Poland 1990–2019

**Jan Banaś** [1,*] **, Katarzyna Utnik-Banaś** [2] **, Stanisław Zięba** [1] **and Krzysztof Janeczko** [3]

[1] Department of Forest Resources Management, University of Agriculture in Kraków, al. 29 Listopada 46, 31-425 Kraków, Poland; stanislaw.zieba@urk.edu.pl

[2] Department of Management and Economics of Enterprises, University of Agriculture in Kraków, al. Mickiewicza 21, 31-120 Kraków, Poland; katarzyna.utnik-banas@urk.edu.pl

[3] Department of Forest Management Planning, Dendrometry and Forests Economics, Institute of Forest Sciences, Warsaw University of Life Sciences—SGGW, Nowoursynowska 159, 02-776 Warsaw, Poland; krzysztof_janeczko@sggw.edu.pl

\* Correspondence: jan.banas@urk.edu.pl; Tel.: +48-696-726-154

**Abstract:** The present work applied a data envelopment analysis (DEA) model to assess changes in the technical efficiency of timber production at a period of transition in forest management priorities. The study material consisted of data on timber sold by Polish State Forests (PSF) and on its forest management inputs in the years 1990–2019. During the period of economic transition in Poland (1990–2003) the technical efficiency of timber production on average amounted to 0.809 and was highly variable. In the free-market period, that efficiency was much higher (on average 0.939) and more stable. This improvement was achieved by substantial layoffs in the PSF, and steady rise in the share and standing volume of mature stands, which made it possible to increase timber production while adhering to sustainable management principles. Analysis of the various categories of inputs to timber production revealed that the greatest decline, in labor costs, was attributable to a fourfold reduction in the workforce, and was accompanied by a 2.4-fold increment in timber production. On the other hand, logging costs increased due to, among other causes, the pursuit of environmentally friendly but more expensive harvesting procedures and reductions in clearcutting, which entailed more dispersed harvesting operations.

**Keywords:** timber production; technical efficiency; forest management costs; data envelopment analysis; multifunctional forestry

## 1. Introduction

While timber has played and continues to play an important role in the economic development of society, the approach to timber harvesting and production has substantially evolved over the years. Early on, the harvested volumes were small compared to the massive forest resources, which were treated as inexhaustible. However, the conversion of increasingly large forest stretches into farmland and the growing demand for timber in the construction sector and in the rapidly expanding industry eventually led to shortages, especially in highly developed regions with cities and industrial centers, intensively cultivated farmlands, and road networks facilitating timber transportation. The problem of timber shortages was addressed by the adoption of planned forest management based on a normal forest model developed in the 19th century by Hundeshagen [1] The model ensured a continuous and stable timber supply with a preference for tree species enabling the fast growth of merchantable timber volume, mainly pine and spruce, harvested under a clearcutting system. This led to proliferation of monospecific and even-aged stands designed to maximize timber production, but significantly hindered non-production (environmental and recreational) forest functions. Extensive monocultures were at high risk of

damage by wind, insects, and fire, which necessitated considerable outlays for protective measures such as chemical insecticides and fire control. When damage did occur, stands were often subjected to salvage logging before reaching rotation age, which entailed considerable economic losses [2]. Yin and Newman [3] analyzed the effect of catastrophic risk on forest investment decisions and stated that the occurrence of catastrophic events always results in a loss of production value.

The progressive environmental degradation and global climate change associated with industrial development have induced calls to adopt new ways of managing natural resources, especially forests, and to respect the general principle of sustainable silviculture defined at the 1993 Ministerial Conference in Helsinki.

In Poland, the legal foundations for environmentally-oriented forestry were laid down by the 1991 Forest Act [4], according to which the production function of forests is treated on an equal footing with the environmental function. The multifunctional forestry model implemented according to that act is founded on a paradigm of sustainable development of ecological, economic, and social forest functions. It should be noted that in addition to revenues from timber production (which are the source of financing for forest management activities), woodland provides other ecosystem services beneficial for humans. At the same time, used wooden elements are biodegradable and do not pose major waste management problems, in contrast to artificial wood substitutes in the construction and furniture industries and other sectors.

This work analyzes how the gradual implementation of sustainable management principles has affected the relative technical efficiency of timber production. Two different approaches to technical efficiency assessment are found in the literature: parametric (stochastic) and non-parametric (deterministic). The parametric approach, also known as stochastic frontier analysis (SFA), is based on frontier production functions [5]. The measure of technical inefficiency is one of the random components of the function, with its other random components consisting of weather disturbances, worker illnesses, and other random factors affecting production [6,7]. The frontier is most often defined by means of Cobb–Douglass (CD) or translog production functions (TL). The parameters of those functions can be used to determine elasticity coefficients, with the elasticities of inputs in the CD function being global across the input space and not dependent on the level of inputs [8,9]. The TL production function is based on flexible functional forms to ensure that elasticities depend on the functional form of the vector of inputs [10].

Non-parametric methods employ data envelopment analysis (DEA), which in contrast to parametric methods does not require assumptions about functional forms. In the DEA method, a deterministic efficiency frontier is the equivalent of a production function, whereas the measure of inefficiency is the distance between the enterprise in question and the frontier representing fully efficient production. The technical efficiency of production in the forest sector is most often analyzed with DEA, rather than SFA [11]. Yin [12] employed DEA to measure the cost efficiency of linerboard producers in North America. Kao et al. [13] used DEA for measuring the efficiency of Taiwan forest management at a time of reorganization [14–16]. The usefulness of DEA for evaluating management efficiency was analyzed by Shiba [17] using the example of Japanese forests. DEA is widely used for measuring eco-efficiency by analyzing both the desirable and undesirable outputs of forest enterprises [18–21]. Within the forest industry, Shasi and Dia [22] used DEA to measure sawmill efficiency in Ontario; it was also employed by Diaz-Balterio et al. [23] to study Spain's wood-based industry. Yin [24] analyzed the efficiency of bleached softwood pulp production in 102 mills around the world using both DEA and SFA methods, finding that in general SFA cost-efficiency levels were higher than their DEA counterparts, and that much of the cost inefficiency was caused by an inappropriate input mix (allocative inefficiency) rather than by the way in which inputs were converted into outputs (technical inefficiency). Kovalčík [25] evaluated the efficiency of forestry in European countries using DEA models adopting labor and timber sale orientations. The same author [26] employed DEA for the efficiency evaluation of forest contractors in mountain and lowland areas. The

operational efficiency of timber harvesting contractors in New Zealand was analyzed by Obi and Visser [27] using a DEA model.

Lundmark et al. [28] integrated the DEA model with the spatial forest sector model to analyze efficiency improvements in the harvesting of forest products. Neykov et al. [29] used the DEA approach and Malmquist productivity index to assess the economic efficiency of forest enterprises in Slovakia and Bulgaria. A similar methodology was used to estimate the technical efficiency of wood-processing companies in Slovakia and Bulgaria and to reveal some factors for efficiency improvements [30]. Młynarski et al. [31,32] carried out an efficiency evaluation of forests districts in Southern Poland using the DEA approach and revealed that lowland forest districts were more efficient than highland ones.

The technical efficiency of timber production by the PSF in the years 1993–1995 was analyzed by Siry and Newman [33] using SFA with a Cobb–Douglas production frontier. The findings indicated substantial technical inefficiency (due inter alia to overstaffing), along with some economies of scale. According to Siry and Newman [33], the PSF's efficiency could be improved by increasing timber production without additional inputs. Alternatively, if timber production could not be significantly increased due to the prevailing market conditions, then the same production levels could be achieved using fewer inputs. Yin [34] introduced modified Faustman–Samuelson–Smith model for assessment of forest investment using pulpwood, chip-n-saw, and sawtimber as outputs, and the opportunity costs of capital (in the form of stocking volume, land rental costs, and operating costs) as inputs:. The results indicated that the predominant cost component is opportunity cost. In contrast, land rental cost and operating cost account for a small percentage of the total cost. The same author [35] analyzed the productivity and profitability of timber production with different management regimes, and stated that by increasing operating inputs, the land base necessary for timber production can be reduced and more land can be used for other purposes such as wildlife habitat, nature reserves, recreational areas or streams, and highway buffers. Profits from timber production in southeastern U.S. private forestry were analyzed by Newman and Wear [36] by employing a restricted profit function of two outputs (sawtimber and pulpwood) and three inputs (regeneration effort, area of forest land, and growing stock).

The effectiveness of Chinese state-owned forest enterprises (SOFEs) has been the subject of several studies. Han et al. [37] measured technical efficiency using DEA to evaluate overall trends and examine how reforms affected the production of social and environmental goods by SOFEs from 2003 to 2009. They reported no overall trend in pure technical efficiency over time for the social firm framework; however, there was an increase in pure technical efficiency for the profit maximization framework and a decrease in scale efficiency, primarily due to higher levels of governmental investment. No evidence was found that forest tenure reform improved technical efficiency. In a study based on a DEA model, Ning et al. [38] attributed the low eco-efficiency of SOFEs in the years 2003–2016 to their low pure technical efficiency. Furthermore, they concluded that eco-efficiency declined in that period due to the implementation of the Natural Forest Protection Project. However, due to a relative lack of production factor inputs, in most SOFEs returns to scale were on an increase. The authors recommended that in the future efforts should be made to promote market-oriented reforms to improve efficiency. Xiong et al. [39], who used stochastic frontier analysis to calculate forestry production efficiency in six provinces of Northwest China, reported that from 2005 to 2015 it gradually decreased. They also found that per capita GDP, forest coverage rate, educational level of forestry employees, and the number of township forestry technology stations were positively correlated with production efficiency. At the same time, collective forest tenure reform had a negative effect on efficiency at the regional level in Northwest China. Yang et al. [40] measured the total factor productivity (TFP) of 135 major SOFEs in 2001–2011 using Malmquist–data envelopment analysis. According to those authors, the technological progress of SOFEs positively affected TFP variation, however technical efficiency increased only slightly

and scale efficiency negatively affected TFP variation. It was also noted that government investments in science and technology accelerated forestry development in China.

Similar to agricultural crops, timber is distinct in that its basic factor of production is land, which is a limited resource characterized by low substitutability with other factors, such as capital or labor. Forest management differs from agriculture in two important respects, i.e., a very long production cycle and the nature of the crop (standing volume is annually increased by timber volume increments; it is a necessary input in the production process, while also being the output). In multifunctional forestry, standing trees fulfill an important biological role as an element of forest ecosystems in addition to their production function. In a multifunctional system, where timber production is one of many forest management goals, increases in environmental expenditures are not directly motivated by economic reasons (greater timber production and lower costs), but rather by the principles of sustainability. Timber production has traditionally been the main objective of forest management [41]. The concept of multifunctionality that arose in the last century [42] obliged forest managers to embrace different management objectives, including non-timber forest products. The sustainability concept in forestry also changed and societies nowadays demand not only a long-term stable flow of timber products but also a stable provision of basic essential environmental goods and services (e.g., reduced soil erosion, biodiversity conservation, carbon sequestration, etc.) [43,44]. In multifunctional forestry, sustainability of forests is measured using multicriteria methods [45–47].

The objective of the present work is to analyze and evaluate the technical efficiency of timber production under changing political and economic circumstances, with an increased focus on non-production forest functions. Relative technical efficiency is determined using two approaches: (1) one output with multiple inputs, analyzing production volume in relation to aggregate inputs in a given year, (2) one output with one input, analyzing separately the effects of individual inputs such as land, labor, and capital (including standing volume, expenditure on logging, silviculture, forest protection, and other, mainly management costs) on timber production volume. The paper is organized as follows: first, a short background of forest management in Poland is given; next, the DEA model is introduced, and data and changes in the technical efficiency of timber production are presented; Finally, the paper ends with concluding remarks.

## 2. Multifunctional Forest Management in Poland

While woodland is multifunctional by its very nature, forest management is oriented towards maximizing selected function(s), depending on the current needs in a given time and space. In Poland, the prioritization of the various forest functions has evolved significantly over the years. Until the 1990s, the predominant function of most stands belonging to Polish State Forests (PSF) was timber production. As a result of the 1991 Forest Act and the adopted international obligations, the PSF adopted a multifunctional forest management orientation based on the principles of sustainable forest development, with a balance between environmental, economic, and social functions. The implementation of multifunctional management objectives was gradual, as reflected by the steady increase in the share of forests designated for predominantly non-production functions.

Over the 30-year study period, the total area of PSF forests grew from 6815 thousand ha in 1990 to 7115 thousand ha in 2019 (Table 1). In that time, the share of protection forests rose significantly from 39.2% to 53.8%. In protection forests, depending on their dominant functions, management practices were modified, e.g., by reduced clearcutting, increased rotation age, altered species composition, the implementation of recreational facilities, etc. In the analyzed period, the share of soil protection forests doubled and that of water protection forests almost tripled, which was also the case with the area of nature reserves.

**Table 1.** Total area and the shares of production and protection forests managed by Polish State Forests in 1990–2019.

| Forest Category | 1990 | 1995 | 2000 | 2005 | 2010 | 2015 | 2019 |
|---|---|---|---|---|---|---|---|
| Total area (ha) | 6815 | 6868 | 6953 | 7042 | 7072 | 7100 | 7115 |
| Production (%) | 60.8 | 52.7 | 51.1 | 53.6 | 52.5 | 47.8 | 46.2 |
| Protection (%) | 39.2 | 47.3 | 48.9 | 46.4 | 47.5 | 52.2 | 53.8 |
| Promotional forest complexes (%) | | 4.9 | 6.4 | 14.1 | 14.2 | 17.9 | 17.9 |
| Nature reserves (%) | 0.5 | 0.6 | 0.9 | 1.1 | 1.2 | 1.4 | 1.5 |
| Natura 2000 | | | | | | 39.0 | 45.4 |

Source: Reports of the Polish State Forests [48].

Of note are the newly defined forest protection categories, such as protection zones (PZs), promotional forest complexes (PFCs), and Natura 2000 sites. PFCs are functional areas of special social, environmental, and educational significance [49]. The first seven PFCs, with a combined area of 333.7 thousand ha, were established in 1994; over the years their number grew to 25 with a total area of 1273.7 thousand ha. They are usually located in large, dense forest complexes representative of local site conditions and forest species composition patterns. In addition to maintaining sustainable silviculture, the main aim of PFCs is to promote forestry and active wildlife protection among the public. That aim is augmented by research designed to comprehensively describe and monitor biocenoses in PFCs as well as by educational activities taking advantage of the developing didactic infrastructure [50,51].

The educational activity is supplemented by the recreation and tourism functions of forests. Forests are made available to the public for those purposes in such a way as to ensure safety to humans and protection of natural resources by the appropriate management of tourism flows and infrastructural development.

Natura 2000 is a form of nature conservation adopted by European Union member states as a means of implementing the provisions of the Birds Directive [52] and the Habitats Directive [53]. It consists of a network of special protection areas (SPAs) for birds and special areas of conservation (SAC) for habitats. Within Natura 2000 sites, protection does not cover all wildlife but is focused on the species and/or habitat types listed in the annexes to the Birds Directive and the Habitats Directive. In Poland, most Natura 2000 sites have been established in woodland areas, which reflects the great significance of forests to biodiversity. In 2019, as much as 38.0% of the overall PSF forest area (2889 thousand ha) was covered by Natura 2000 sites [54].

The forest management certification process in Poland began in 1996 [55,56]. It was initiated by the Regional Directorates of State Forests at the request of the exporters of timber products, for whom having a certificate was a requirement demanded by buyers and which was later used as a marketing tool. In 2020, whole forest areas managed by State Forests were PEFC (Pan-Europe Forest Certification) certified and nearly 90% of State Forests were FSC (Forest Stewardship Council) certified [57].

## 3. Data and Methods

### 3.1. Theoretical Background for Efficiency Measurement

Efficiency is the ratio of outputs to the inputs involved in a given production process. Technical efficiency was defined by Koopmans [58] as follows: a producer is technically efficient if an increase in any output requires a reduction in at least one other output or an increase in at least one input, and if a reduction in any input requires an increase in at least one other input or a reduction in at least one output. Thus, a technically inefficient producer could produce the same outputs with less of at least one input, or could use the same inputs to produce more of at least one output [59]. Debreu [60] and Farrell [61] introduced a measure of technical efficiency defined as one minus the maximum equiproportionate reduction in all inputs that still allows continued production of given outputs. Farrell proposed that the efficiency of a firm consists of two components: technical efficiency, reflecting its ability to obtain maximal outputs from a given set of inputs, and allocative

efficiency, reflecting its ability to use the inputs in optimal proportions given their respective prices and production technology. These two measures are then combined to provide a measure of total economic efficiency.

Our study measured efficiency by means of data envelopment analysis (DEA), a nonparametric method that uses linear programming to construct a nonparametric piecewise surface (or frontier) over the data [62]. Such a piecewise-linear convex isoquant combines all points represented by fully efficient firms. Efficiency measures are then calculated relative to this surface. Comprehensive reviews of DEA methodology have been presented by Charnes et al. [63], Cooper et al. [64], Färe and Lovell [65], and Petersen [66].

Here a firm or enterprise, known in the DEA literature as a decision-making unit (DMU), is defined as an entity transforming inputs into outputs. A change in the technical efficiency of a DMU means that it moves closer to or further away from the frontier.

DEA models may be classified into input-oriented and output-oriented, and can be further subdivided into those adopting constant returns to scale (CRS) and variable returns to scale (VRS) [67]. In input-oriented models, technical inefficiencies (DMU distance to the frontier) indicate a possible proportional reduction in input utilization at a constant output level. Conversely, in output-oriented models, technical inefficiencies indicate a possible increase in outputs with the input level held constant. The technical efficiency levels obtained from input- and output-oriented models are the same under CRS, but not VRS. The choice of an appropriate orientation is not as important in the case of linear programming (DEA) as it is in the case of econometric estimation (SFA), which suffers from a simultaneous equation bias.

### 3.2. Characteristics of the DEA Model

To build the DEA model for a timber production process, let $x = (x_1, x_2, \ldots, x_i)$ be a vector of $i$ inputs and $y = (y_1, y_2, \ldots, y_j)$ be a vector of $j$ outputs of $n$ DMU, where $n = 1, \ldots, N$. In this study DMU corresponds to consecutive calendar years. The best way to introduce DEA is via the ratio form. For each DMU we would like to obtain a measure of the ratio of all outputs over all inputs such as:

$$\theta_n = \frac{\sum_j u_{jn} \, y_{jn}}{\sum_i v_{in} \, x_{in}} \tag{1}$$

where $\theta_n$ is the technical efficiency score of timber production in year $n$ while $u$ and $v$ are weights for each output and input, respectively. The key problem with computing $\theta$ is constructing the input and output weights. For this purpose, Charnes et al. [68] proposed a linear programming formulation known as the CCR model, assuming a technology that exhibits constant return to scale (CRS).

The input-oriented BCC model proposed by Banker et al. [69] assumes variable returns to scale (VRS) and evaluates the efficiency of $DMU_0$ by solving the following linear program:

$$
\begin{aligned}
&\min_{\theta \lambda} \theta \\
&\text{subject to :} \\
&\sum_n x_{in} \lambda_n \leq \theta x_{i0}, \ i = 1, 2, \ldots, i \\
&\sum_n y_{jn} \lambda_n \geq y_{j0}, \ j = 1, 2, \ldots, j \\
&\text{(i) } \sum_n \lambda_n = 1; n = 1, \ldots, n \\
&\lambda_n \geq 0
\end{aligned}
\tag{2}
$$

The optimal solution vector for the above problem will contain the technical efficiency scores $0 < \theta^* \leq 1$. The BCC model requires two-step optimization. In the first step, $\theta$ is minimized using Equation (2), and in the second step, the sum of input surpluses and output shortfalls, known as slacks [70], is maximized while holding constant $\theta^*$ (optimal objective value obtained in step one). Another way of treating slacks, applied in this work, is multi-step DEA, where a sequence of radial linear programming problems is used to identify the efficient projected point. [71]

A scale efficiency measure can be obtained by performing both CRS and VRS DEA. The technical efficiency scores obtained from CRS DEA can be decomposed into a scale efficiency component and a "pure" technical efficiency component [72]. If the CRS and VRS technical efficiency scores for a given firm differ, this indicates that it suffers from scale inefficiency. In that case, scale efficiency (SE) can be calculated as the ratio of the $TE_{CRS}$ $T_{VRS}$ and $TE_{CRS}$ scores [73]:

$$SE = TE_{CRS}/TE_{VRS} \tag{3}$$

When SE < 1, the firm can behave in two ways regarding returns to scale, exhibiting either increasing returns to scale when outputs grow proportionately faster than inputs, or decreasing returns to scale when outputs grow more slowly than inputs.

In this study, the efficiency scores given by Equations (1)–(3) were calculated using DEAP 2.1 software [67].

*3.3. Material*

The study material consists of data on the volume of timber sold by Polish State Forests (PSF) as well as its forestry management inputs in the years 1990–2019. Given the scope of the information and the long study period, the data were obtained from the following sources: (1) the volume of timber sold, standing volume, and area of forested land are from annual PSF management reports (1990–2004) and from the State Forest Information System (2005–2019); (2) the number of full-time PSF employees, expenditures on logging (harvesting and skidding), silviculture, forest protection, and other costs are from annual PSF financial reports. The "other costs" category consists of expenditures on multifunctional forestry, including the development and maintenance of a database of forest resources under all forms of ownership linked to data on wildlife protection and the quality of the natural environment, forest condition monitoring based on the National Forest Inventory (NFI), and research and technological work to improve the implementation of nonproduction forest functions.

Timber production amounted on average to 29.6 ($\pm$7.8 standard deviations) million m$^3$ year$^{-1}$, and increased 2.4-fold from 17.5 million m$^3$ in 1990 to 41.1 million m$^3$ in 2019 (Figure 1a). The total area of forests managed by the PSF was on average 6.70 million ha, and grew incrementally at a mean rate of 0.01 million ha per year (Figure 1b). Standing volume steadily increased from 1262 billion m$^3$ in 1990 to 2066 billion m$^3$ in 2019 (Figure 1b). In the first half of the analyzed period, the number of PSF employees decreased almost fourfold, from 104.3 thousand in 1990 to 26.5 thousand in 2004, while after 2004 it remained relatively stable at around 25.7 thousand (Figure 1c). The mean expenditures on forest management were \$927.45 million, of which 41.1% were logging costs, 16.4% were silviculture costs, 9.4% were forest protection costs, and 33.1% were other costs (Figure 1d).

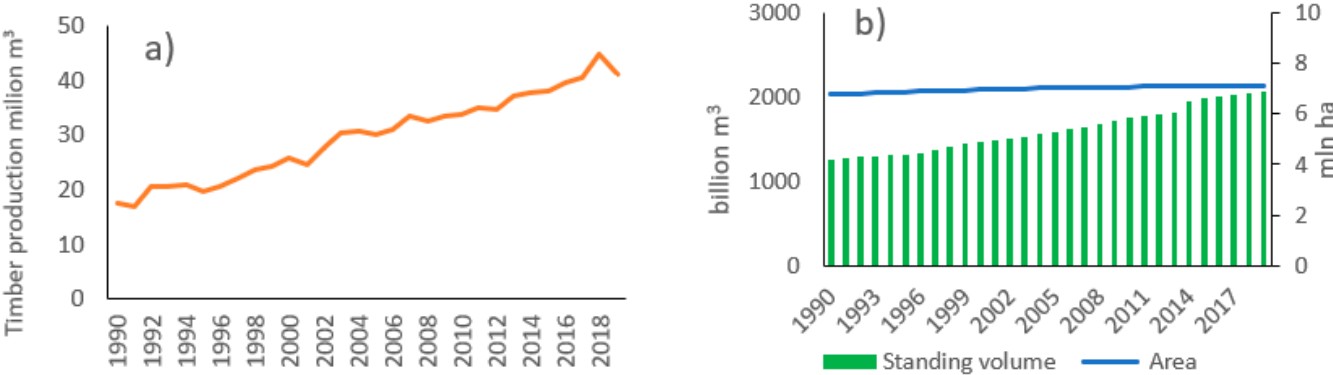

**Figure 1.** *Cont.*

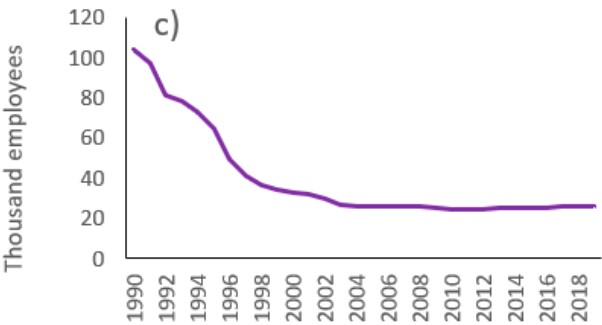 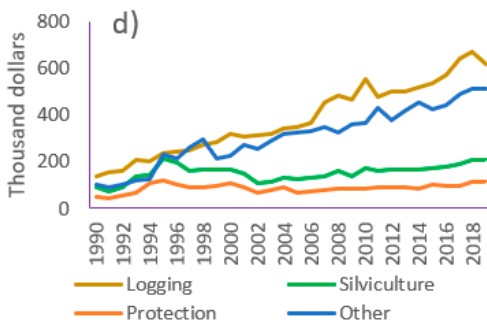

**Figure 1.** Output (**a**) and different inputs of timber production: (**b**) standing volume (left axis) and forested area (right axis); (**c**) labor costs; (**d**) costs of logging, silviculture, forest protection, and other costs.

Inputs per 1 m³ of timber produced in the years 1990–2019 are given in Table 2. In general, unit costs of logging and other costs increased approximately twofold, while there was a decrease in unit inputs of forest land (twofold), standing volume (0.4-fold), and labor (ninefold). Unit silviculture and forest protection costs were highly variable in the first decade of study, after which they stabilized at their initial levels.

**Table 2.** Inputs per 1 m³ of timber produced by the Polish State Forests in 1990–2019.

| Inputs | 1990 | 1995 | 2000 | 2005 | 2010 | 2015 | 2019 |
|---|---|---|---|---|---|---|---|
| Land (ha) | 0.39 | 0.35 | 0.27 | 0.23 | 0.21 | 0.19 | 0.17 |
| Standing volume m³ 10³ | 72.2 | 67.40 | 56.70 | 52.84 | 51.82 | 52.09 | 50.30 |
| Labor (10³ employees) | 5.97 | 3.30 | 1.28 | 0.87 | 0.73 | 0.67 | 0.64 |
| Logging costs (USD) | 7.90 | 12.13 | 12.34 | 11.59 | 16.37 | 14.15 | 15.05 |
| Silviculture costs (USD) | 5.02 | 10.94 | 6.37 | 4.19 | 5.08 | 4.56 | 5.06 |
| Protection costs (USD) | 2.75 | 6.17 | 4.19 | 2.30 | 2.57 | 2.76 | 2.73 |
| Other costs (USD) | 5.98 | 11.76 | 8.68 | 10.88 | 10.74 | 11.09 | 12.46 |

## 4. Results

### 4.1. Changes in Technical Efficiency

The relative technical efficiency of timber production in the years 1990–2019 amounted on average to 0.885 (Figure 2). Analysis of technical efficiency scores revealed two distinct periods, corresponding to the economic transition between 1990 and 2003 and the free-market economy period between 2004 and 2019 (beginning with Poland's accession to the EU in 2004). While in the former period mean technical efficiency was 0.809 and highly variable (0.110 standard deviation), in the latter period it was much higher (on average 0.939) and less variable (0.032 standard deviation).

In the years 1992, 2003, 2013 and 2018 the PSF operated at an optimal scale (TE = 1.000). In the remaining years the PSF operated at increasing returns to scale, which means that timber production increased at a greater rate than inputs. A mean inefficiency equal to 0.115 (1–0.885) indicates that in the analyzed period, the volume of timber produced could have been greater by 11.5% without any additional inputs.

### 4.2. Productivity of Inputs

Land productivity amounted on average to 0.671, increasing steadily from 0.408 in 1990 to 1.000 in 2018 (Table 3, Figure 3a). Throughout the study period the pure technical efficiency of land remained similar at about 0.850, with increasing returns to scale; consequently, the increase in timber production (2.5-fold) was greater than that in land efficiency (1.3-fold).

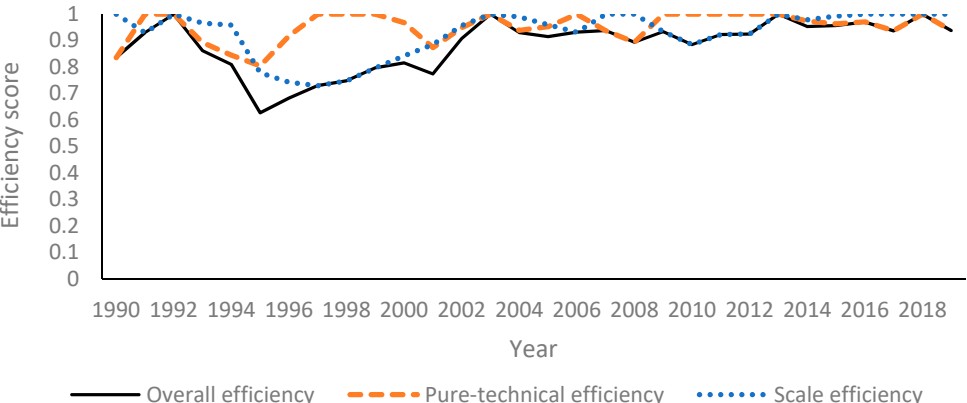

**Figure 2.** Technical efficiency of timber production by Polish State Forests in the years 1990–2019 according to the one output multiple inputs model.

**Table 3.** Technical efficiency of timber production by Polish State Forests in 1990–2019 according to the one output one input approach.

| Input | Overall Efficiency | Pure Technical Efficiency | Scale Efficiency | Returns to Scale * |
|---|---|---|---|---|
| Land | 0.671 (0.168) | 0.833 | 0.803 | irs |
| Standing volume | 0.829 (0.099) | 0.940 | 0.883 | irs |
| Labor | 0.584 (0.295) | 0.712 | 0.763 | drs/irs |
| Logging costs | 0.659 (0.133) | 0.921 | 0.718 | drs |
| Silviculture costs | 0.792 (0.174) | 0.840 | 0.941 | drs |
| Forest protection | 0.795 (0.184) | 0.823 | 0.962 | drs |
| Others costs | 0.548 (0.173) | 0.911 | 0.601 | drs |

* irs—increasing returns to scale, drs—decreasing returns to scale, standard deviations are given in brackets.

Standing volume productivity (Figure 3b) amounted to an average of 0.829 (0.099) at high pure technical efficiency throughout the period (mean of 0.940). In this case, scale efficiency grew from 0.635 in 1990 to 0.911 in 2003, and remained at a high level (above 0.920) after 2004. Labor productivity (Figure 3c) exhibited the greatest variation among the analyzed inputs throughout the study period, with an average score of 0.584. It increased significantly from 0.099 in 1990 to 0.664 in 2003, with decreasing returns to scale; after 2004 it grew at a much slower rate with increasing returns to scale. The productivity of logging inputs (Figure 3d) amounted to an average of 0.659, steadily declining from 1.000 in 1990 to 0.524 in 2019. In 1990–1992, the PSF exhibited an optimal scale of logging inputs, but since 1993 it operated at decreasing returns to scale, which means that logging inputs were increasing faster than the volume of timber produced.

The productivity of silviculture (Figure 3e) and forest protection inputs (Figure 3f) were similar both in terms of their value and direction of change over time. In 1990–1991 they exhibited high efficiency, then declined quickly to reach the lowest point in 1995 (0.395 and 0.369, respectively). Subsequently, efficiency increased at a steady pace up to 2003, after which it remained stable at approx. 0.900. The productivity of the remaining forestry inputs (Figure 3g) was on average 0.909 in 1990–1994, but in 1995 it dropped precipitously to 0.435. Subsequently, after a small increase in 2000, the efficiency of those inputs exhibited a slight downward trend up to the end of the study period.

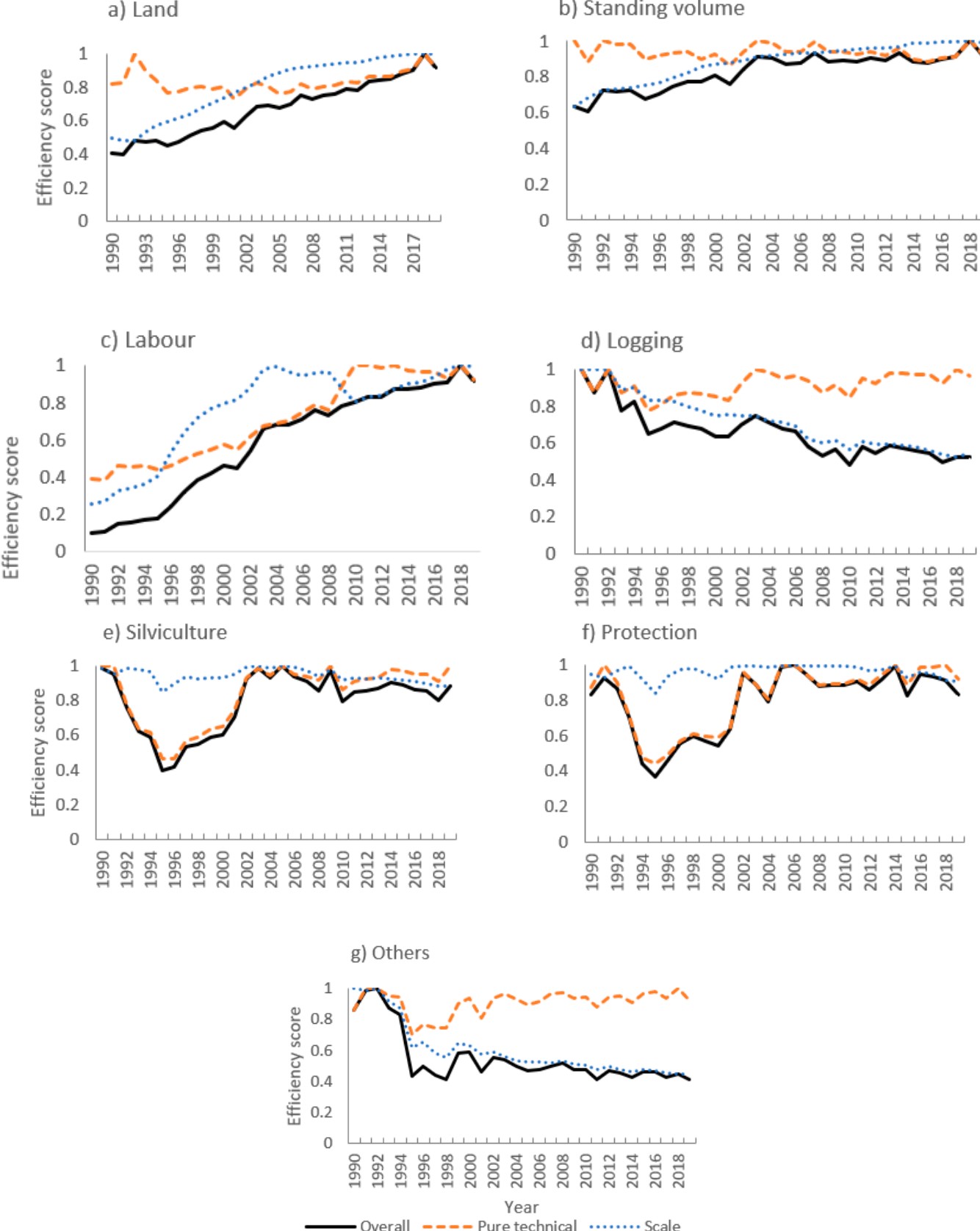

**Figure 3.** Technical efficiencies of inputs to timber production by State Forests in Poland in 1990–2019.

## 5. Discussion

This study applied the DEA method to evaluate the efficiency of timber production in Polish State Forests during Poland's economic transition and subsequently under a free market system. The relative technical efficiency scores calculated for individual years well reflect both the economic changes and the shift in forest management from a production-oriented paradigm towards a multifunctional one. The technical efficiency of timber production by the PSF during the economic transition period (1990–2003) initially declined, reaching the lowest point in 1995 (0.627). The drop in efficiency was mostly due to surging labor costs (not only in forestry, but also in industry, construction, and agriculture). Thus, to increase its efficiency, the PSF substantially reduced its workforce, from 104.34 thousand employees in 1990 to 26.9 thousand in 2003. The layoffs mostly involved blue-collar workers, with many jobs requiring manual labor being outsourced to external companies via public tenders.

Under the command economy, the timber production process did not reflect the market forces of supply and demand. Both outputs (timber quantity and prices) and inputs (wages and the prices of production factors) were fixed. Forestry was focused on maximizing timber production, whose levels were determined by multi-year plans. The relative technical efficiency of timber production was high.

The economic transition led to the deregulation of the prices of inputs and outputs, which initially caused a rapid drop in the technical efficiency of timber production (labor costs soared while timber production levels remained at similar levels). In 1995, the overall operating costs of the PSF exceeded its revenues, resulting in losses. Over time, the production efficiency of the PSF improved as a result of (1) substantial workforce reduction, (2) a transition towards market-oriented timber sales [74,75] (not analyzed in this work), and (3) a continuous increase in mature stands, enabling greater production volumes while adhering to the principles of sustainable forestry.

Kao [14] applied a DEA method to assess the performance of Taiwanese forests following reorganization. Similarly, as in this work, the inputs examined by Kao were land, labor, expenditures (money spent each year), and initial forest stock volume (from before the evaluated period). The studied outputs were timber production (as in this work), soil conservation expressed by forest stock (higher stock volumes lead to lower soil erosion), and recreation (defined as the number of visitors served by forests every year). A major purpose of forest reorganization in Taiwan was to improve economies of scale. Over three years, workforce reduction amounted to 60.9%, expenditures decreased by 9.7%, and the initial stock increased by about 10%; despite this, no significant overall improvement was observed from the viewpoint of technology or scale, with the average scores remaining almost identical (0.893 in 1989 vs. 0.894 in 1992).

In this work, the one output one input approach was used to determine the partial efficiencies of individual inputs, revealing some marked differences in terms of both their values and directions of change over time. The underlying causes of change in the efficiency of individual inputs are as follows: the increased productivity of forest land and standing volume is attributable to a much greater increase in timber production (2.4-fold) as compared to that of land (4.4%) and standing volume (63.7%) throughout the study period; the improvement in labor efficiency resulted from a fourfold workforce reduction and an increase in timber production; the efficiency of forest protection and silviculture was improved by the implementation of sustainable forestry principles (e.g., the conversion of monocultures), leading to greater stand stability and lower costs of silviculture and forest protection (e.g., controlling insect outbreaks and fires); the share of natural regeneration (less expensive than artificial regeneration) grew from 4.2% in 1990 to 13.7% in 2019 [48]; the decreased technical efficiency of logging is associated with soaring labor costs, implementation of environmentally friendly but costly harvesting systems, and reduced clearcutting with greater dispersion of harvesting operations.

The lower productivity of the remaining inputs is attributable to the gradual growth of that group of costs, including expenditures on nonproduction forest functions involving

tourist and recreation infrastructure, forest education, wildlife conservation, monitoring of Natura 2000 sites, the functioning of forest promotional complexes, and the promotion of multifunctional forestry in the social media.

## 6. Conclusions

Analysis of the technical efficiency of timber production revealed two distinct periods: the economic transition 1990–2003 and the free-market period 2004–2019 (after Poland joined the European Union in 2004). Under the command economy, timber production was not subjected to the market forces of supply and demand. Both outputs (timber volume and prices) and inputs (wages and the prices of production factors) were fixed. In the course of economic transition, the prices of the inputs and outputs were deregulated, initially causing a precipitous decline in the technical efficiency of timber production due to soaring production costs, especially wages, while production volume remained at similar levels. In 1995, the overall costs of the PSF's operations exceeded its revenues, generating negative profits.

The efficiency of timber production improved over time as a result of (1) considerable layoffs in the PSF, (2) a market reorientation of the timber sales strategy, and (3) a steady rise in the share and standing volume of mature stands, which enabled greater production volumes while respecting sustainable management principles.

The shift in forest management from a production model to a multifunctional one led to an increase in forest areas designated for protection and social functions, where the intensity of timber production was gradually reduced by departing from clearcutting, extending rotation age, replacing monocultures with multi-species uneven-aged stands, and the use of retention trees, up to the complete discontinuation of harvesting in selected stands. Analysis of the various inputs to timber production revealed that the greatest decrease was in labor costs, caused both by a fourfold workforce reduction and a 2.4-fold increase in timber production. On the other hand, logging expenditures grew due to the implementation of more expensive environmentally friendly harvesting methods and the limited use of clearcutting, which entailed more fragmented harvesting operations. Finally, there was a marked increase in expenditures related to the implementation and development of non-production forest functions.

**Author Contributions:** Conceptualization, J.B.; Methodology, J.B. and K.U.-B.; Software, S.Z.; Validation, K.U.-B. and K.J.; Investigation, J.B. and S.Z.; Writing—Original draft preparation, J.B. and K.U.-B.; Writing—Review and editing, J.B. and K.U.-B.; Visualization, J.B. and K.U.-B.; Supervision, J.B. and K.U.-B. All authors have read and agreed to the published version of the manuscript.

**Funding:** This study was financed by The Ministry of Science and Higher Education of the Republic of Poland.

**Institutional Review Board Statement:** Not applicable.

**Informed Consent Statement:** Not applicable.

**Acknowledgments:** We would like to thank the State Forests National Forest Holding in Warsaw for making data available to us. We also thank the three anonymous reviewers and editor for the effective feedback provided.

**Conflicts of Interest:** The authors declare no conflict of interest.

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
