# Peer review of "Assessing the Technical Efficiency of Timber Production during the Transition from a Production-Oriented Management Model to a Multifunctional One: A Case from Poland 1990–2019"

_forests, doi:10.3390/f12091287_

Round 1

Reviewer 1 Report

The particular study examines the efficiency of timber production in Poland during the period 1990-2019. It is also provides details about the big changes that the shift to the economy brought to the sustainability of the national forest resources. The article seems interesting and well presented by the authors. I made some suggestions as sticky notes. Please follow them. Important issue is to consider my note at discussion regarding a large part which in my opinion belongs to the introduction.

Author Response

Thank you very much for your time and comments helpful in improving the quality of the manuscript. We have revised the text in terms of “vague and unfounded statements” in an attempt to provide clearer and more accurate information.

We also substantially improved the introduction and discussion adding new relevant references and improved the design.

Reviewer 2 Report

Dear authors,

Thank you for the possibility to review to paper and thereby get a glimpse regarding to the Poland efficiency of timber production during the transition from a production-oriented management model to a multifunctional one between 1990–2019, for forest management priorities. The topic is really current and important.

I think this article has good potential and some aspects need to be clarified and largely improved.

Still, in my opinion the article has a quite limited view on various categories of inputs to timber production, approaching a few domains….. and how they have contributed to sustainable management principles.

 I would recommend you to broaden the literature review (1) and also make the positioning of the paper clearer already at the introduction.

Pleas also make the positioning of the paper clearer already at the final remarks (6). Moreover I am missing more answers to “so what questions” and implications.

English language and style are minor spell check required.

Author Response

(The authors gave the same response as above.)

Reviewer 3 Report

The research is very interesting and topical. The study of the economic efficiency of forestry using non-parametric methods is a problem that requires a lot of analysis and discussion. In this respect, the study is well targeted.
It should be noted that the analysis of problems lacks an in-depth citation and analysis of the use of DEA for the needs of the forestry sector in Europe. Citing Kovalchik (2018) is good, but more is needed. There are very recent new studies on this issue in the journal Forest. The research methodology is very close to these studies and they must find a place in the analyzes. The methodology also offers a combination of methods that is completely identical to other studies in the field of forestry, forest industry and the circular economy in the forest sector - it is good to cite these sources. Input and output-oriented models are also described in the analyzes. What does require their joint use? The formula (3) has not been use anywhere. It is necessary to explain the usage or input-oriented model, or the output-oriented  one. The resulst are not clear on how have they been produced - with output-oriented or input-oriented model. 

In conclusion, the research can be improved to good one if it is based on  the previous studies tretty close to this one.  By this way the research will present its uniqueness and value to the forestry not only in Poland. 

I wish you fruitful work

Author Response

(The authors gave the same response as above.)

Round 2

Reviewer 2 Report

Dear author

Thanks to the authors for their efforts in revising the manuscript. I recommend publishing the article in its present form.

Author Response

Thank you for your time and the effective feedback provided.

Reviewer 3 Report

The manuscript has been significantly improved. It is now in the sufficient form to be published in a journal like Forests. Good job.

Author Response

(The authors gave the same response as above.)
